# Astrocyte- and Neuron-Derived Extracellular Vesicles from Alzheimer’s Disease Patients Effect Complement-Mediated Neurotoxicity

**DOI:** 10.3390/cells9071618

**Published:** 2020-07-04

**Authors:** Carlos J. Nogueras-Ortiz, Vasiliki Mahairaki, Francheska Delgado-Peraza, Debamitra Das, Konstantinos Avgerinos, Erden Eren, Matthew Hentschel, Edward J. Goetzl, Mark P. Mattson, Dimitrios Kapogiannis

**Affiliations:** 1Laboratory of Clinical Investigation, Intramural Research Program, National Institute on Aging, National Institutes of Health (NIA/NIH), Baltimore, MD 21224, USA; carlos.nogueras-ortiz@nih.gov (C.J.N.-O.); francheska.delgado-peraza@nih.gov (F.D.-P.); konstantinos.avgerinos@nih.gov (K.A.); erden.eren@nih.gov (E.E.); matt.hentschel@emory.edu (M.H.); 2Department of Neurology, Johns Hopkins School of Medicine, Baltimore, MD 21205, USA; vmachai1@jhmi.edu (V.M.); ddas5@jhmi.edu (D.D.); 3Department of Medicine, University of California, San Francisco, CA 94143, USA; edward.goetzl@ucsf.edu; 4San Francisco Campus for Jewish Living, San Francisco, CA 94112, USA; 5Department of Neuroscience, Johns Hopkins School of Medicine, Baltimore, MD 21205, USA; mmattso2@jhmi.edu

**Keywords:** neurodegeneration, exosomes, astrocytes, complement, membrane attack complex

## Abstract

We have previously shown that blood astrocytic-origin extracellular vesicles (AEVs) from Alzheimer’s disease (AD) patients contain high complement levels. To test the hypothesis that circulating EVs from AD patients can induce complement-mediated neurotoxicity involving Membrane Attack Complex (MAC) formation, we assessed the effects of immunocaptured AEVs (using anti-GLAST antibody), in comparison with neuronal-origin (N)EVs (using anti-L1CAM antibody), and nonspecific CD81+ EVs (using anti-CD81 antibody), from the plasma of AD, frontotemporal lobar degeneration (FTLD), and control participants. AEVs (and, less effectively, NEVs) of AD participants induced Membrane Attack Complex (MAC) expression on recipient neurons (by immunohistochemistry), membrane disruption (by EthD-1 assay), reduced neurite density (by Tuj-1 immunohistochemistry), and decreased cell viability (by MTT assay) in rat cortical neurons and human iPSC-derived neurons. Demonstration of decreased cell viability was replicated in a separate cohort of autopsy-confirmed AD patients. These effects were not produced by CD81+ EVs from AD participants or AEVs/NEVs from FTLD or control participants, and were suppressed by the MAC inhibitor CD59 and other complement inhibitors. Our results support the stated hypothesis and should motivate future studies on the roles of neuronal MAC deposition and AEV/NEV uptake, as effectors of neurodegeneration in AD.

## 1. Introduction

Alzheimer’s disease (AD) is the result of a neurodegenerative cascade involving progressive deposition of misfolded amyloid beta-peptide (Aβ) and tau, abnormalities in brain cell homeostasis and function, and ineffective or even maladaptive compensatory mechanisms [1]. Recently, neuroinflammation and its cellular mediators, microglia and astrocytes, have emerged as important factors in AD pathogenesis [2] and have been implicated in the development of both Aβ [3,4] and tau [5] pathologies. Maladaptive neuroinflammation in AD involves the complement cascade [6,7], a system of sequentially activated humoral and cellular proteins that lies on the interface between innate and adaptive immunity and promotes host defenses against infection and tissue homeostasis. The complement cascade is classically known to be activated by IgM or IgG immunocomplexes (classical pathway), bacteria or toxins (alternative pathway) or mannose residues (lectin pathway) [8], ultimately leading to the formation of the membrane attack complex (MAC), a cytolytic membrane pore formed by the sequential assembly of soluble complement proteins C5b, C6, C7, C8, and C9, that causes cell osmolysis after intercalation into the plasma membrane [9,10]. For peripheral tissues, the main source of complement proteins is the liver, whereas the brain relies on local synthesis observed in all cell types [11,12,13]. Appropriately regulated complement is necessary for brain defense and homeostasis, but an overactivated complement might lead to brain pathology.

Multiple lines of evidence implicate the complement cascade in AD pathogenesis. In AD mouse models, accumulation of oligomeric Aβ and hyperphosphorylated tau leads to the glial overexpression of several complement components, and molecular inhibition or genetic deletion of complement factors/receptors, such as C3/C3R, C5/C5R, and C1q decrease glial activation, Aβ and hyperphosphorylated tau deposition, and synaptic pruning [7,14,15,16,17]. Complement components of the classical pathway colocalize with Aβ plaques and tau neurofibrillary tangles in the hippocampus, and the temporal and frontal lobes of AD patients [13,18,19,20]. In addition, the MAC was detected in the vicinity of Aβ and tau deposits, both in the inner and outer leaflets of the neuronal plasma membrane [21]. Furthermore, genome wide association studies (GWAS) associated CR1 (gene encoding the receptor for C3b/4b) with late onset AD [22]; the interaction between C3b and CR1 mediates Aβ phagocytosis [23]; thus, polymorphisms altering the CR1 function could decrease Aβ clearance.

Extracellular vesicles (EVs) are membranous nanoparticles produced by all cells, including neurons and astrocytes. The role of EVs in the brain is multifaceted and includes the exchange of cargo molecules between neurons and glia, to regulate neuronal activity, but also the transfer of misfolded α-synuclein, Aβ, and hyperphosphorylated tau, possibly contributing to neurodegenerative disease propagation [24,25,26,27]. Taking advantage of the blood–brain barrier permeability to EVs from the brain to the periphery [28], we and others have isolated neuronal and astrocytic origin-enriched EVs (NEVs and AEVs, respectively) from plasma and human cells, through particle precipitation, followed by immunocapture with anti-GLAST and anti-L1CAM antibodies, respectively, and showed that they are carriers of AD pathogenic proteins [29,30,31]. Previously, we showed that AEVs of AD patients carry high levels of multiple complement components of the classical and alternative pathways (i.e., C1q, C4b, C3b, Bb, C3d, factor B, and factor D), as well as the MAC [32]. Interestingly, we also found levels of endogenous complement regulatory proteins, such as CD59, that binds to the transmembrane residues C8 and C9 and blocks the formation of the MAC [9,10], CD46, and the decay-accelerating factor (DAF), to be low in EVs from AD patients [32]. These findings motivate the hypothesis that astrocytic and neuronal EVs of patients with AD could transmit complement proteins to neurons, upon their uptake, resulting in MAC formation and MAC-associated neurotoxicity. In this study, we utilize NEVs and AEVs immunoprecipitated from the plasma of individuals with AD or controls, as a tool for investigating the molecular mechanisms in vitro that might reflect the pathogenic events occurring in the brain in AD. Specifically, to provide proof-of-principle for neurotoxicity mediated by AEV and AEV uptake and MAC formation, we conducted in vitro assays that showed that circulating AEVs and NEVs of AD patients could be uptaken by neurons and that they exert neurotoxicity through mechanisms involving MAC deposition, membrane disruption, and necroptosis.

## 2. Materials and Methods

### 2.1. Human Subjects

All participants donated blood as part of their participation in the clinical studies at the National Institute on Aging (NIA), approved by the National Institutes of Health Institutional Review Board or the Johns Hopkins Institutional Review Board. All participants provided written informed consent. Procedures for sample collection and processing were identical for all samples. NIA participants included 7 participants with early AD and 6 age- and sex-matched cognitively normal controls, randomly selected from the cohort used for the original report of high AEV complement in AD [32], as well as 2 participants with FTLD (one with behavioral variant frontotemporal dementia and one with semantic dementia) (Appendix A). Since this study aimed to investigate mechanisms pertaining to AD, only plasma samples of subjects with additional proof for harboring AD pathology besides clinical diagnosis were included.

Individuals with AD whose baseline plasma samples were used in this study were participants in a clinical trial of exenatide in AD (https://clinicaltrials.gov/ct2/show/NCT01255163). As noted in the published results from this study [33], the eligibility criteria included age > 60 years, clinical diagnosis of amnestic Mild Cognitive Impairment (MCI) or probable AD (at the mild stage), CSF Aβ42< 192 pg/mL (using INNO-BIA Alz Bio3 kits), and absence of other neurological disorders or significant neuroimaging abnormalities. All participants had a clinical dementia rating (CDR) global score of 0.5 (corresponding to MCI) or 1 (corresponding to mild dementia). From this clinical trial cohort and for the purpose of this study, we selected the baseline samples (i.e., before randomization) of individuals that fulfill the criteria for high probability AD, based on clinical diagnosis of amnestic MCI or probable AD, low CSF Aβ42, high CSF total tau, or p181-tau, although their enrollment preceded the development of recent biomarker-based diagnostic criteria [34,35]. Moreover, upon review of individual cases, the individuals used for this study qualify as Amyloid + /Tau + /Neurodegeneration + according to the A/T/N framework [36]. As stated in the publication of the clinical study, “the levels (pg/mL) of Aβ42, p181-tau and total tau were determined at the University of Pennsylvania Biomarker Research Laboratory using Luminex ×MAP technology with INNO-BIA Alz Bio3 kits provided by Fujirebio” [33]. Of note, this is the same laboratory that conducted the biomarker measurements for Alzheimer’s Diseases Neuroimaging Initiative (ADNI) using the same settings and kits. Therefore, there was a direct comparability of the CSF values with ADNI participants with MCI or AD [37]. The FTLD participants met the clinical criteria [38]. Normal controls were participants at the Baltimore Longitudinal Study on Aging conducted at the NIA, who remained cognitively normal for the course of their participation; they had Blessed Information Memory Concentration Test score < 4, and CDR = 0. To replicate the key findings of the AEV-mediated MTT cell viability decrease we used additional plasma samples from 13 JH ADC participants with autopsy-confirmed AD and 3 neurologically normal controls (no demographic or additional information was provided on these subjects). Limited quantities of these samples precluded us from using them in additional functional experiments. EVs isolated from individual subjects were not pooled and their effects were assessed and analyzed separately, to respect their biological variability.

### 2.2. Cell Cultures

Primary cortical neurons were derived from Sprague Dawley^®^ rats at embryonic day 18, isolated from timed pregnant animals (Charles River Laboratories, Frederick, MD) and maintained as described previously [39]. All procedures were approved by the National Institute on Aging Animal Care and Use Committee and complied with the NIH guidelines. In brief, the concentration of dissociated neurons in B27-supplemented neurobasal medium (Thermo Fisher Scientific, Waltham, MA) was determined using an Hemocytometer, and the cells were cultured in polyethylenimine-coated plates or coverglass in a cell culture incubator, at 37 °C and 5% CO_2_, for 14–21 days in vitro (DIV), prior to the experiments.

### 2.3. Ngn2-Mediated Neuronal Differentiation of Human iPSCs

Human-induced pluripotent stem cells (iPSCs) were maintained as feeder-free cells in Essential 8 medium on vitronectin-coated plates (all from Gibco, Carlsbad, CA, USA) [40]. The medium was changed every day and the cells were passaged at 80–90% confluency, using TrypLE Express Enzyme (Gibco) supplemented with 10 μM Y-27632 (Stem Cell Technologies, Vancouver, Canada). The human iPSC line used in this study was the BC1 reprogrammed from peripheral blood mononuclear cells of a healthy male collected, under a Johns Hopkins Institutional Review Board-approved clinical study [40].

The protocol followed for the neuronal differentiation of human iPSCs was adopted from Zhang et al. [41] who reported that forced expression of the single transcription factor Ngn2 could convert iPSCs into functional neurons with very high yield, in less than 3 weeks. In brief, human BC1 iPSCs were treated with Accutase (Innovative Cell Technologies, San Diego, CA, USA) and plated as dissociated cells (25 × 10^4^ cells/well) on day 2 in a 6-well Matrigel-coated plate (BD Biosciences, San Jose, CA, USA). On day 1, Ngn2 lentiviral infection (2.5 µg/mL; CHOP Research Vector Core, Philadelphia, PA, USA) was performed using Polybrene (1 μg/μL; Sigma Aldrich, St. Louis, MO, USA). The virus-infected cells were expanded, and frozen stocks were made for future differentiation experiments. *TetO/NGN2* gene expression was induced by Doxycycline (2 µg/mL; Clontech, Madison, WI, USA) on day 0. Puromycin (2.5 µg/mL) was added to the medium on day 1 for 24 h. The surviving cells were harvested on day 2 and plated on a Matrigel-coated 24 well plate at a concentration of 1 × 10^5^ cells/well. The cells were fed with neural differentiation media containing B27, BDNF (10 ng/mL), NT3 (10 ng/mL), every other day until day 12. Cells were treated with 2 µM cytosine β-d-arabinofuranoside hydrochloride (Ara-C; Sigma Aldrich) on day 4, to reduce the proliferation of non-neuronal cells. Doxycycline was discontinued after day 12 and the cells were fed every two days thereafter until day 21, when the neurons were mature enough to harvest.

### 2.4. Isolation of Astrocyte- and Neuronal-Derived Extracellular Vesicles from Human Plasma

All blood draws were conducted between 7 and 10 am and after an overnight fast at the NIA Clinical Unit, following standard procedures. Approximately 10 mL of venous blood were collected in plasma separator tubes containing EDTA, incubated for 10 min at room temperature (RT) and then centrifuged at 3000 rpm for 15 min at RT. Supernatant plasma was divided into 0.5 mL aliquots and stored at −80 °C, until further use. Hemolysis was ruled out using spectrophotometry (data not shown). Pre-analytical factors for blood collection and storage complied with the guidelines for EV biomarkers [42,43].

Plasma samples were thawed on ice and immediately subjected to isolation of AEVs and NEVs, using a methodology extensively described elsewhere [29,30,32]. In brief, fibrinogen, a coagulating protein highly abundant in plasma and thought to impede efficient EV recovery, was removed using 5 U/mL of thrombin (System Biosciences, Inc.; Mountainview, CA, USA) for 30 min at RT, followed by addition of 495 µL of Dulbecco’s PBS-1X (DPBS), supplemented with protease (cOmplete^TM^ Protease Inhibitor Cocktail; Millipore Sigma, Burlington, MA, USA) and phosphatase inhibitors (Halt^TM^ Phosphatase Inhibitor Cocktail; Thermo Fisher Scientific) and centrifugation at 6000× *g* for 20 min at 4 °C. The supernatant was transferred to a sterile 1.5 mL microtube and total EVs were sedimented by incubation with 252 µL of ExoQuick^®^ (System Biosciences)—a proprietary solution that allowed the sedimentation of EVs from low volumes of plasma and other biofluids, without the need for ultracentrifugation—for 1 h at 4 °C, followed by centrifugation at 1500× *g* for 20 min at 4 °C. The EV-depleted supernatant was transferred to a sterile 1.5 mL microtube and stored in −80°C. Pelleted total EVs were resuspended by overnight gentle rotation mixing at 4 °C in 500 μL of DPBS supplemented with protease and phosphatase inhibitors. Resuspended total EVs were incubated with 4 μg of anti-GLAST (i.e., antibody against the astrocyte cell surface antigen-1; RRID: AB_2733473; Miltenyi Biotec, Auburn, CA, USA), anti-human CD171 (i.e., antibody against neural cell adhesion molecule L1CAM; RRID: AB_2043813; Thermo Fisher Scientific), or anti-human tetraspanin CD81 (Ancell Corporation, Bayport, MN, USA) biotinylated antibodies, to immunocapture AEVs, NEVs, and endosome-derived CD81 + EVs of variable cell origin [44], respectively, for 4 h at RT. EV-antibody complexes were incubated with 30 µL of washed Pierce™ Streptavidin Plus UltraLink™ Resin (Thermo Fisher Scientific), for 1 h at RT; EV-antibody-bead complexes were allowed to sediment by gravity, followed by removal of unbound EVs and soluble proteins in the supernatant. Bound AEVs, NEVs, and CD81+ EVs were eluted using 200 µL of 0.1 M glycine (stock solution at 1 M, pH = 2.7; Polysciences Inc., Warrington, PA), followed by the immediate transfer of the supernatant to a sterile microtube containing 20 µL of 1 M tris buffer (pH = 8), for pH neutralization.

Ten microliters of intact EVs were used for determination of particle concentration and diameter, using nanoparticle tracking analysis (NTA) (Nanosight NS500; Malvern, Amesbury, UK). The remaining 210 µL of intact EVs was aliquoted into sterile microtubes in 20 µL aliquots, or mixed with 1.5 parts of Mammalian Protein Extraction Reagent (MPER) lysis buffer (Thermo Fisher Scientific), supplemented with protease and phosphatase inhibitors for downstream immunoassays, and then stored at −80 °C until further use.

The size and morphology of the intact AEVs were assessed using Transmission Electron Microscopy. EVs were absorbed on to a 400-mesh carbon-coated copper grid (cat. no. CF400-Cu; Electron Microscopy Sciences, Hatfield, PA, USA) for 1 min, and then quickly rinsed using ddH_2_O. Then, EVs were stained with 2% uranyl acetate (Electron Microscopy Sciences). All imaging was done on a Zeiss Libra 120 (Zeiss, Thornwood, NY, USA) with an Olympus Veleta camera (Olympus America, Center Valley, PA, USA).

### 2.5. Characterization of AEVs and NEVs by Immunoblotting and ELISA

To confirm the sequential purification of EVs from neat plasma using ExoQuick sedimentation of total EVs followed by immunoprecipitation of AEVs, NEVs, and CD81 + EVs, we assessed positive and negative EV protein markers in EV-depleted plasma, total EV isolates, and immunoprecipitated AEVs, NEVs, and CD81 + EVs, using immunoblotting, according to the established criteria [45]. The protein concentration was determined using the Bradford protein assay (Biorad, Hercules, CA, USA). One microgram of total protein per sample was resolved by sodium dodecyl sulfate polyacrylamide gel electrophoresis (SDS-PAGE) using 4–12% Bis-Tris gels in MOPS SDS running buffer at 200 V, for 50 min (NuPAGE^®^ Novex^®^ SDS-PAGE system; Thermo Fisher Scientific) and transferred to polyvinylidene fluoride membranes, using the iBlot^®^ 2 gel transfer system (Thermo Fisher Scientific) programmed to run at 20 V for 1 min, 23 V for 4 min, and 25 V for 6 min. Membranes were blocked using the Odyssey^®^ blocking buffer in tris buffer saline (TBS; Licor Biosciences, Lincoln, NE, USA) for 1 h at RT and incubated overnight at 4 °C with fluorescently labelled antibodies (IRDye^®^ 680RD antibody infrared dye; Licor Biosciences) targeting apolipoprotein A1 (0.4 µg/mL; RRID: AB_2242717, cat. no. AF3664, R&D Systems, Minneapolis, MN, USA) as an indicator of lipoproteins, the cis-Golgi marker GM-130 (0.2 µg/mL; RRID: AB_880266, cat. no. 52649, Abcam, Cambridge, MA, USA) used as a negative EV marker, and the membrane and intra-vesicular positive EV markers CD81 (1:500; cat. no. EXOAB-CD81A-1, System Biosciences) and alix (0.4 µg/mL; RRID: AB_11023702, cat. no. NBP1-90201, Novus Biologicals, Littleton, CO, USA), respectively [45]. Antibody excess was washed five times with TBS supplemented with 0.05% tween-20 detergent for 5 min and the blots were scanned using the Odyssey^®^ CLx imaging system (Licor Biosciences).

To further characterize our EV preparations, we measured the protein levels of common trans-membrane EV markers, tetraspanins, CD63, CD81, and CD9, on intact NEV and AEVs, using a multiplex electrochemiluminescence-based ELISA assay developed by Meso Scale Discovery (Meso Scale Discovery, Rockville, MD, USA).

### 2.6. Neurotoxicity Assays

The effects of AEVs, NEVs, CD81+, and total EVs, from normal control, AD, and FTLD participants, on the metabolic activity of primary neurons were evaluated using the tetrazolium salt 3-(4,5-dimethylthiazol-2-yl)-2,5-diphenyltetrazolium bromide (MTT) assay (Trevigen, Gaithersburg, MD, USA), a colorimetric test based on the enzymatic activity of NADPH-dependent cytoplasmic oxidoreductase enzymes that catalyze the reduction of the membrane permeable MTT into colorimetric formazan products. Additionally, EV effects on neuronal membrane integrity were assessed using the membrane impermeable dye ethidium homodimer (EthD-1) (Cell Biolabs, San Diego, CA, USA), which fluoresces when bound to DNA.

To ensure that the effects of EVs used as treatments to recipient neurons were not attributed to differences in EV concentration, we used the NTA EV concentration data to adjust and fix the dose of EVs provided with each treatment (by diluting as needed). Moreover, we further fixed the dose of EVs provided with each treatment per number of plated cells (i.e., 100,000 cells per well of a 96-well plate). The EV concentration used to treat neurons and treatment duration were based on a previous study, in which the neurotoxicity of CSF EVs of AD patients was evaluated using E18 rat cortical neurons [46], and confirmed by experiments assessing the optimum concentration- and time-dependency of neurotoxicity of immunoprecipitated EVs.

E18 rat cortical neurons cultured for 14–21 DIV in PEI-coated 96-well plates at a confluency of 100,000 neurons per well, in 200 µL of neurobasal media were treated with AEVs, NEVs, CD81+, and total EVs of AD, FTD, and the normal participants, at 50, 100, and 600 EVs/neuron in 20 µL of neurobasal media for 24, 48, and 72 h, in triplicates. These were then subjected to MTT and EthD-1 assays, following the manufacturers’ instructions. Neuronal culture in corner wells of the plate was avoided and 300 µL of ddH_2_O was added instead to prevent artificial alterations in cell viability caused by medium evaporation, a phenomenon known as the ‘edge effect’. Treatments with 10–100 µM glutamate for 24 h and 0.1% saponin detergent (Cell Biolabs; [47,48]) for 10 min were used as positive controls of neurotoxicity, whereas neurons treated with 20 µL of neurobasal media (vehicle treatment), were used as a negative control. Plates were read using the Synergy^TM^ H1 plate reader and Gen5^TM^ software (BioTek Instruments). Given that the fluorescent signal from the EthD-1 plates was emitted from adherent cells unevenly distributed at the bottom of the wells, the surface area of each well was scanned using the 9 × 9 area scan read module and the average signal was used for data analysis. MTT and EthD-1 signals from the EV-treated samples and positive controls of neurotoxicity were normalized to vehicle treatment, prior to statistical analysis. To evaluate if the neurotoxicity of AEVs and NEVs on the MTT assay depended upon MAC formation, the MTT assay was carried with co-treatment with CD59. Primary neurons and EVs were pre-incubated with active recombinant human CD59 (cat. no. 1987-CD, R&D Systems), at a final concentration of 1 µM (stock at 10 µM) for 2 h in the cell culture incubator, prior to the treatment of cells with EVs, to allow cells to incorporate CD59 in the plasma membrane, as described previously [9].

The complement-mediated neurotoxicity of AD AEVs was further validated in a human-based model, using human iPSC-derived neurons cultured in Matrigel-coated 12-well plates for 12–18 DIV at a cell confluency of 100,000 neurons per well, in a final volume of 500 µL of neural differentiation media. The cell confluency and plate selection were based on the optimal conditions for cell differentiation and growth. The EthD-1 membrane disruption assay was carried out using human iPSC-derived neurons treated with 800 EVs/neuron in 35 µL of neurobasal medium (final concentration of EVs in culture: 1.5 × 10^8^ EVs/mL) for 48 h, with and without the addition of inhibitors of the classical pathway (Factor I and CR1), alternative pathway (Factor H and DAF), and terminal pathway (CD59), as described above. The authors aimed to replicate the EV treatment conditions mediating neurotoxicity in rat cortical neurons, by keeping the final EV concentration consistent (2.7 × 10^8^ EVs/mL; 600 EVs/cell for 100,000 cells in a final volume of 220 µL). However, the amount of EVs required to achieve such conditions was unfeasible, and approximately half of the concentration was used instead (1.5 × 10^8^ EVs/mL; 800 EVs/cell for 100,000 cells in a final volume of 500 µL of media plus 35 µL of diluted EVs). The concentration used for each inhibitor was determined using the reported EC_50_ values, as follows: Factor I, 70 nM, stock at 2 µM; CR1, 10 nM, stock at 1.88 µM; Factor H, 235 nM, stock at 10 µM; DAF, 70 nM, stock at 2 µM; CD59, 500 nM, stock at 10 µM. CR1 (cat. no. 5748-CD), Factor H (cat. no. 4779-FH), DAF (cat. no. 2009-CD) and CD59 (cat. no. 1987-CD) were acquired from R&D Systems, whereas Factor I (cat. no. 00003426-Q01) was acquired from Novus Biologicals. Information regarding the determination of the EC50 for each complement inhibitor was provided by the suppliers. EthD-1 results acquired using the fluorescence plate reader were further validated qualitatively by visualization of EthD-1 positive nuclei, using an Olympus IX50 inverted microscope, equipped with a U-LH100HG fluorescence light source (Olympus America) and an AxioCam HrC camera (Zeiss).

### 2.7. Assessment of Neurite Fragmentation by Immunocytochemistry

The neurite density of EV-treated rat cortical neurons was assessed by means of β III-tubulin (neuron-specific class III tubulin, also known as Tuj-1) fluorescent immunoreactivity, to evaluate whether EV treatments induce neurite fragmentation [49]. Neurons cultured in PEI-coated 8-well chambered borosilicate coverglass (Thermo Fisher Scientific) at a confluency of 200,000 neurons/well in 500 µL of neurobasal media were treated in duplicate with 20 µL of AEVs, NEVs, and CD81 + EVs from AD and normal participants in neurobasal media at 600 EVs/neuron (final concentration of EVs in culture: 2.4 × 10^8^ EVs/mL), for 48 h, using 100 µM glutamate treatment as a positive control of neurotoxicity. After treatment with EVs, neurons were fixed with 4% FA, followed by membrane permeabilization with 0.1% triton X-100 detergent in DPBS, for 15 min at room temperature. Excess detergent was washed three times with DPBS and non-specific antibody binding was blocked with DPBS supplemented with 2% bovine serum albumin, for 1 h at RT. Then, the cells were incubated with 1 µg/mL of mouse anti-β III-tubulin antibody (RRID: AB_2256751, cat. no. 78078, Abcam) in blocking solution, overnight at 4 °C, followed by five washes with DPBS and incubation with 1 µg/mL Alexa Fluor^TM^ 647 donkey anti-mouse antibody (RRID: AB_162542, cat. no. A-31571, Thermo Fisher Scientific) for 1 h at RT. Excess of secondary fluorescent antibody was washed five times with DPBS and ProLong^TM^ Diamond Antifade Mountant, with the nucleic acid stain DAPI (cat. no. P36962, Thermo Fisher Scientific) added prior to the visualization of cells, using fluorescence confocal microscopy. Ten to fifteen images per sample were acquired at 10X and 25X magnifications and analyzed using the image processing software Fiji [50]. The ‘moments’ threshold image filter was applied and the surface image area of β III-tubulin was selected, measured, and normalized to the amount of DAPI+ nuclei, counted using the ImageJ Image-based Tool for Counting Nuclei (ITCN) plugin (Center for Bio-Image Informatics, University of California, Santa Barbara, CA, USA).

MAC immunocytochemistry was conducted in the EV-treated rat cortical neurons to evaluate MAC expression in recipient cells. Neurons were treated with AEVs from AD and normal participants, for 1 h, and then subjected to fluorescent immunolabeling for MAC, using a mouse anti-human C5b-9 antibody (1 µg/mL; RRID: AB_2067162, cat. no. M0777, Dako, Carpinteria, CA, USA) targeting an epitope in the poly-C9 component of MAC, in association with C5b, a protein quaternary structure characterizing the active complex [51], and showing cross-reactivity with the C5b-9 equivalent protein in rat [52]. To determine the neuronal localization of MAC immunoreactivity, cells were co-labeled with the neuronal marker Tuj-1 [53] (rabbit anti-β III-tubulin antibody, RRID: AB_444319, cat. no. 18207, Abcam), at 1 µg/mL using Alexa Fluor^TM^ 488 goat anti-mouse and 647 goat anti-rabbit (RRID: AB_2534069, cat. no. A11001; and RRID: AB_2535813, cat. no. A21245; Thermo Fisher Scientific) as secondary antibodies. MAC and β III-tubulin immunoreactivity were visualized via fluorescence confocal microscopy and 5–10 images per sample were acquired at 10, 20, and 63X magnifications. Images were analyzed using the image processing software Fiji and the mean fluorescence of MAC positive regions of interest was selected using the ‘moments’ threshold image filter determined.

### 2.8. Assessment of EV-Mediated Activation of Apoptosis and Necroptosis

To determine if apoptosis or necroptosis underlie the neurotoxicity of AD AEVs, E18 rat cortical neurons cultured at a density of 1.6 × 10^6^ neurons/well in PEI-coated 12-well plates were treated with 100 µL of AEVs from AD and normal participants in neurobasal media at a final concentration of 600 EVs/neuron (9.6 × 10^8^ EVs/mL), for 48 h in triplicates. After incubation, the culture media was removed and neurons were washed once with DPBS, followed by cell lysis using 50 µL of RIPA buffer (Thermo Fisher Scientific), supplemented with phosphatase and protease inhibitors. The protein concentration of pooled triplicates was determined using the BCA protein assay (Thermo Fisher Scientific) and 8 µg of the total protein subjected to immunoblotting against cleaved caspase-3 and -7 (1:200 dilution; RRID: AB_2341188 and RRID: AB_11178377; cat. no. 9661S and 8438S, Cell Signaling, Danvers, MA, USA), as well as phosphorylated (1:500 dilution; cat. no. MABC1158, Millipore Sigma) and total MLKL (1:500 dilution; RRID: AB_2737025, cat. no. 172868, Abcam), using β-actin (RRID: AB_306371, cat. no. 8226, Abcam) as the loading control, using 10% bis-tris gels in MES SDS running buffer.

Activation of apoptosis by EVs was confirmed using an in-vivo fluorescence microscopy assay designed to detect active caspases in live cells (Image-iT^TM^ Live Green Poly Caspases Detection Kit, cat. no. I35104, Thermo Fisher Scientific). This assay is based on the detection of the membrane permeable active caspase inhibitor FLICA^TM^. E18 rat cortical neurons cultured for 14–21 DIV in PEI-coated 8-well chambered cover glasses at a confluency of 100 neurons/well in 500 µL of neurobasal media were treated with 20 µL of AEVs and NEVs in neurobasal media at 600 EVs/neuron in duplicates, for 48 h using 0.1% saponin as a positive control for caspase activation [47]. After treatment with EVs, neurons were subjected to the in vivo poly-caspases detection assay, according to the manufacturer’s instructions, and visualized under a fluorescence confocal microscope. 3–10 images per sample were acquired at 10X magnification and the mean fluorescence of FLICA positive regions of interest was determined using the Fiji’s ‘max entropy’ threshold image filter. Signals from EV- and saponin-treated samples were normalized to vehicle treatment and subjected to statistical analysis.

### 2.9. Experimental Design and Statistical Analyses

Analyses were performed using SPSS version 25. For analyses of biological effects of EV treatments, we used mixed linear models for repeated measures to assess the fixed effects of Group (i.e., AD, normal, and FTLD), on vehicle-normalized values of MTT, EthD-1, and neurite density assays. To appropriately handle the biological and technical variability, respectively, we analyzed individual measurements, modeling the correlated random effects for Subject and using Well (from 96-well plates of cultured neurons) and Plate as repeated measures (15 well repetitions on average per EV type per Subject). For significant effects of Group, we conducted t-tests of least square means to determine the direction and significance of pairwise differences (e.g., comparing effects of AEVs from AD vs. normal participants); reported comparisons were Bonferroni-corrected. Alternative models including participant age and sex did not result in better fit (by AIC and BIC criteria). Mixed models for repeated measures were also used to analyze MTT and EthD-1 rescue experiments, assessing the effect of Condition (AEVs alone, AEVs plus CD59, AEVs plus Factor H, AEVs plus Factor I, AEVs plus DAF). Models comparing EV types (AEVs, NEVs, CD81+, and total EVs), included terms for EV type and the Group*EV type interaction. Comparison of results from additional experiments (i.e., concentration- and time-dependent EV treatments, C5b-9 accumulation and activation of apoptosis and necroptosis in EV-recipient neurons, CD59 protein levels in EVs using ELISA, and neurotoxicity of trypsinized vs. intact EVs) were made using two-tailed unpaired t-tests or ordinary one-way ANOVA corrected for multiple comparisons, using the Dunnet test. Statistical details of the experiments can be found in the figure legends and results.

## 3. Results

### 3.1. Characterization of Plasma-Derived EVs

According to the established criteria [45], we characterized our EV preparations by determining the size and concentration of isolated EVs by Nanoparticle Tracking Analysis (NTA) (Appendix A) and Transmission Electron Microscopy (Appendix A). The size distribution of nanoparticles immunocaptured by L1CAM, GLAST, or CD81 ranged from 50 to 250 nm, with a particle diameter mode of 100 nm, and was consistent with a mixed EV population likely predominated by exosomes (50–100 nm) and microvesicles (100–200 nm), whereas, the total EVs showed higher diameters up to 400 nm (Appendix A). EV purification of immunoprecipitated EVs was confirmed by Western blots, showing the enrichment for transmembrane and intravesicular EV markers (i.e., CD81 and alix), low levels of apolipoprotein A1 and absence of the Golgi-specific negative EV marker GM-130, in comparison with EV-depleted plasma and total EVs, as previously seen [31] (Appendix A). AEVs immunoprecipitated from AD and the control subjects carry comparable amounts of CD9, CD63, and CD81 (Appendix A), suggesting that the neurotoxicity of circulating AEVs from the AD subjects cannot be attributed to differences in general EV markers or EV concentration.

### 3.2. AD AEVs and NEVs are Neurotoxic

EVs from individual subjects were not pooled and their effects were assessed and analyzed separately, to respect their biological variability. First, we sought to determine whether AEVs labeled with the fluorescent lipid analogue PKH26 might be internalized by cultured E18 rat cortical neurons (Appendix A), as previously shown for NEVs [46]. After 1 h of incubation, we detected puncta-like PKH26+ structures in the soma and neurites of cells treated with AEVs of AD and normal participants (Appendix A). Then, we used the MTT cell viability assay to evaluate whether AEVs and NEVs from participants with AD (*n* = 7) compared to cognitively normal controls (*n* = 6), might be neurotoxic in a concentration- and time-dependent manner (Figure 1). Both AEVs and NEVs from AD compared to the control participants decreased cell viability, with maximum effects produced by a concentration of 600 EVs/neuron (Figure 1a) and incubation for 48 h (Figure 1b), treatment conditions that were kept constant in subsequent studies, unless otherwise indicated. The neurotoxicity of AD AEVs and NEVs was comparable to excitotoxicity by 10 μM glutamate, hence physiologically relevant [54], and was cell of origin-specific (astrocytic and neuronal) and independent of putative soluble co-precipitates, since no cell viability differences were observed in the neurons treated with nonspecific CD81+ and total EVs from the same AD participants (Figure 1c). In a model including all EV types and terms for Group and Group*EV type interaction, NEVs of AD participants produced a significant MTT decrease (AD vs. Normal, *p* = 0.003), and so did AEVs (AD vs. Normal, *p* = 0.006), but CD81+EVs did not (AD vs. Normal, *p* = 0.904) and neither did total EVs (AD vs. Normal, *p* = 0.675). For the normal controls, there were no significant differences between EV types and no effects on MTT. For the AD participants, NEVs and AEVs had similar effects on MTT (AEVs vs. NEVs, *p* = 0.695), but AEVs decreased MTT compared to both CD81+ (*p* = 0.035) and total EVs (*p* = 0.029), and NEVs showed similar strong trends in pairwise comparisons (*p* = 0.087 and *p* = 0.067 respectively). Results were similar for models including age and sex. Moreover, there were no effects on the viability of neurons treated with AEVs from 2 participants with frontotemporal lobar degeneration (FTLD), compared to 6 controls (*p* = 0.934), suggesting that AEV neurotoxicity is not a common feature of neurodegenerative proteinopathies (Figure 1d). To replicate the AEV-mediated MTT cell viability decrease in a different cohort, we treated rat cortical neurons with plasma AEVs from 13 individuals from the Johns Hopkins Alzheimer’s Disease Center (JH ADC), with autopsy-confirmed AD or neurologically normal controls. AEVs of confirmed AD individuals induced a significant cell viability decrease compared to the vehicle-treated neurons (*p* = 0.02) and neurons treated with AEVs from controls (*p* = 0.02) (Figure 1e).

We used Tuj1 fluorescent immunocytochemistry to further characterize the neurotoxicity of AEV and NEV treatments on recipient neurons, and assess potential neurodegeneration. Tuj1 is a class III β-tubulin antibody that might be used to provide the structural details for axons and dendrites of both developing and mature neurons. In addition to being a marker of neuronal differentiation in primary neuroblasts and developing brains, Tuj1 was found to be useful in the in vitro and in vivo immunochemical detection of neurodegeneration, which was characterized by microtubule disruption and aberrant sprouting of axons and dendrites, and was induced by a variety of stressors, including oligomeric Aβ_42_ and accumulation of Aβ plaques [49]. Results showed a significant neurite density decrease in neurons treated with AEVs from AD, compared to the control participants (*p* = 0.02); the difference was not observed in the NEV-treated (*p* = 0.891) or CD81 + EV-treated neurons (*p* = 0.252) (Figure 2), suggesting a greater neurodegenerative potential for AEVs, compared to NEVs or CD81 + EVs.

### 3.3. Neurotoxicity of AD AEVs is Associated with Complement MAC Accumulation, Membrane Disruption, and Necroptosis Activation

Given our previous finding of high levels of complement effectors, including MAC, in circulating AEVs of AD participants [32], we sought to evaluate if treatment with AD AEVs results in MAC deposition and membrane disruption in recipient neurons. First, we performed MAC fluorescent immunocytochemistry in AEV-treated neurons, using a validated C5b-9 antibody targeting the C5b-C9 heterodimer, the rate-limiting step of MAC pore formation [10], with β III-tubulin co-staining to visualize the neuronal structures. Results showed a dramatic increase of MAC accumulation co-localizing with β III-tubulin in the soma and neurites of cells treated with AD AEVs, for 1 h, compared to neurons treated with AEVs from control participants and vehicle (AD vs. Normal, *p* = 0.026; AD vs. vehicle, *p* = 0.012; Normal vs. vehicle, *p* = 0.051) (Figure 3a,b). To assess whether MAC accumulation leads to plasma membrane disruption, the first step towards osmolysis, we subjected EV-treated cells to the fluorescent ethidium homodimer-1 (EthD-1) membrane disruption assay, which is based on the detection of the membrane impermeable probe EthD-1, which fluoresces in association with nucleic acids (Figure 3c). Treatment of neurons with NEVs and AEVs from 4 AD, compared to 4 control participants, resulted in a significantly increased EthD-1 fluorescence fold-change over vehicle (for NEVs, AD vs. Normal, *p* = 0.044; for AEVs, AD vs. Normal, *p* = 0.024), suggesting that both AD NEVs and AEVs mediate neuronal membrane disruption; there were no differences in EthD-1 fluorescence over vehicle in the neurons treated with CD81+ EVs (*p* = 0.332) from AD patients, compared to the control participants. MAC-dependent cytotoxicity is accompanied by activation of the regulated necrosis pathway, known as necroptosis, which involves phosphorylation of the mixed-lineage kinase domain-like protein (MLKL) [55,56,57]. To further explore the mechanism of neurotoxicity induced by AEVs and NEVs, we evaluated whether EV-treated neurons exhibit necroptosis and apoptosis cell-death markers. We did not observe activation of apoptosis, as suggested by a fluorescent polycaspase in vivo assay that did not show activation of caspases in EV-treated neurons (Figure 4a,b). This was confirmed by Western blots showing that treatment of neurons with AD AEVs did not result in increased levels of the apoptosis markers cleaved caspase-3 and -7, compared to treatment with AEVs from the control participants (Figure 4c,d). Interestingly, neurons treated with AD AEVs showed increased levels of the necroptosis marker phosphorylated MLKL, in comparison to control AEVs (Figure 4c,d), providing evidence for necroptosis activation that is consistent with, although not pathognomonic of, a mechanism of MAC-mediated neurotoxicity.

Next, we replicated our previous finding of low levels of CD59 (a GPI-anchored cell membrane glycoprotein that inhibits MAC assembly and thus protects cells from lysis [9,10]), in circulating AEVs of AD participants [32] (Appendix A). Altogether, these findings promote the hypothesis that the neurotoxicity of AD AEVs and NEVs is due to the MAC-mediated osmolysis, in an environment impoverished of endogenous complement inhibitors, especially CD59.

### 3.4. Complement Inhibition Blocks the Neurotoxicity of AD AEVs and NEVs

As an initial test of the hypothesis that AD AEVs cause neurotoxicity and membrane disruption due to their surface cargo of complement components, we stripped AEVs of their surface protein epitopes by trypsination, which abolished their effect on neuronal viability (Appendix A). Next, to assess whether MAC assembly in situ on the plasma membrane underlies the neurotoxicity of AD AEVs and NEVs, we co-treated rat cortical neurons with CD59 and AEVs or NEVs from AD and control participants and found that CD59 abrogated the AD AEV- and NEV-mediated cell viability decrease on the MTT assay (Figure 5a). (For NEVs, AD vs. Normal, *p* = 0.013; AD plus CD59 vs. Normal, *p* = 0.859; AD vs. AD plus CD59, *p* = 0.113; for AEVs, AD vs. Normal, *p* = 0.001; AD plus CD59 vs. Normal, *p* = 0.609; AD vs. AD plus CD59, *p* = 0.016.) This was further validated in a model of human iPSC-derived neurons subjected to the EthD-1 assay, the most stringent and relevant assay to the mechanism of complement and MAC-mediated cytotoxicity, in which co-treatment with CD59 blocked the AD AEV-mediated membrane disruption (AD vs. AD plus CD59 AEVs, *p* = 0.009; AD plus CD59 vs. Normal AEVs, *p* = 0.863) (Figure 5b,c). Similar, but weaker, protective effects were produced by other endogenous inhibitors that bind and destabilize the C3 and C5 convertases upstream from MAC assembly—complement receptor 1 (CR1; also known as CD35), which inhibits the classical cascade (AD vs. AD plus CR1, *p* = 0.012; AD plus CR1 vs. Normal AEVs, *p* = 0.923), and Factor H (AD vs. AD plus Factor H, *p* = 0.011; AD plus Factor H vs. Normal AEVs, *p* = 0.698), and at trend-level the decay-accelerating factor (DAF; also known as CD55) (AD vs. AD plus DAF, *p* = 0.091; AD plus DAF vs. Normal AEVs, *p* = 0.414), which inhibit the alternative cascade. Importantly, the effects of co-treatment with AD AEVs and only Factor I, which can degrade specific complement effectors only when aided by co-factors (e.g., Factor H and CR1) [8], were not different from the effects of AD AEVs alone (AD vs. AD plus Factor I, *p* = 0.349; AD plus Factor I vs. Normal AEVs, *p* = 0.053).

## 4. Discussion

We demonstrated that AEVs and NEVs circulating in the blood of AD patients could be readily internalized by neurons, induce MAC deposition on their surface, cause membrane disruption and neurite fragmentation, and decrease cell viability. These effects were not produced by nonspecific CD81+ and total plasma EVs from AD patients or AEVs and NEVs from FTLD or cognitively normal participants and were blocked by the MAC inhibitor CD59 and additional endogenous complement inhibitors, suggesting that neuronal MAC assembly represents a necessary effector for AEV and NEV-mediated neurotoxicity in AD. Whereas previous studies have established the trans-neuronal spread of Aβ and tau via EVs [5,58,59], our study contributed to this growing field by demonstrating that EVs can also mediate the propagation of neuroinflammatory mediators in AD. These findings suggest a novel mechanism for neurodegeneration in AD, which involves neuronal internalization of AEVs and NEVs containing high levels of toxic complement components and low levels of endogenous inhibitors and recycling of their cargo to the plasma membrane, leading to in situ complement activation and MAC assembly. As inflammatory astrocytes have much higher levels of complement proteins and complexes than neurons, it is possible that some of the complement of NEVs is derived from AEVs that have entered neurons and then NEVs prior to re-secretion. An additional, albeit less likely, possibility, given the protective effects of pre-treatment with CD59 and other complement inhibitors, is that preformed MAC (which we have shown to be present in AEVs [32]) might be transferred to the plasma membrane of recipient neurons during EV fusion.

These conclusions were supported by employing multiple conditions and controls. To rule out an indiscriminate neurotoxic effect of circulating AEVs and NEVs, AD patients were compared to cognitively normal controls. To show neurodegenerative disease specificity for AD, we assessed FTLD AEV effects. To show specificity for neuronal and astrocytic EV origin, and to rule out the involvement of some putative soluble factor contaminating our EV preparations, we tested multiple circulating EV types from the same subjects, isolated through similar immunocapture procedures, specifically, L1CAM+ (neuronal), GLAST+ (astrocytic), CD81+ (variable cell types), as well as total EVs, showing that only neuronal and astrocytic EVs were neurotoxic. To assess the magnitude of EV-mediated toxicity, we showed it to be comparable to neurotoxicity by 10 μM glutamate and sub-nanomolar oligomeric Aβ_42_, therefore, to also be physiologically relevant [54,60]. By showing that toxicity is abolished by EV trypsinization, we confirmed that it was mediated by surface proteins, such as molecular anchors that mediate EV uptake by recipient neurons [61] or complement components. To rule out that the human-origin of AD EVs might be responsible for toxicity to rat cortical neurons and confirm that identified mechanisms apply to human neurons, we also assessed human iPSC-derived neurons. Whereas the confluency achieved by human iPSC-derived neurons did not allow for the assessment of toxicity with some of the assays conducted on rat cortical neurons (such as the MTT neuronal viability assay), we replicated the key findings of induced membrane disruption and its rescue by complement inhibitors, using the EthD-1 assay. Moreover, the human subjects, from whom EVs were derived, were carefully selected (patients had a high probability for AD based on CSF biomarkers [34] or met the clinical criteria for two types of FTLD; the controls were followed longitudinally over many years and did not show cognitive decline). Studying human circulating EVs rather than culture-derived EVs (such as in [62,63]) rendered greater validity to the findings. EVs from individual subjects were not pooled to avoid the possibility that neurotoxicity could be attributable to a single subject and to respect their biological variability. Since our primary focus was to investigate mechanisms, biological variability was recognized and respected in the analysis, but not exhaustively studied. Finally, the association between the neurotoxicity of AD AEVs and increased complement cargo was demonstrated in an independent cohort involving “gold standard” participants with autopsy-confirmed AD.

In this study, we had to rely on a limited number of subjects with a high availability of plasma, and prioritized the following parameters: immunoprecipitate multiple sub-populations of EVs (NEVs, AEVs, CD81+ EVs) from the same human samples; use EVs from the same human samples for technical replicates for each model system; use EVs from the same human samples for different experimental models (e.g., EVs from the same subjects were used to treat rat and human neurons). This prioritization of experimental controls and consistency in the human biological material utilized was made to ensure convergent validity of results from various assays and model systems, at the expense of fully assessing biological variability. Nevertheless, MTT results were replicated in a second cohort involving autopsy-confirmed patients with AD. Future experiments will aim to assess these effects in larger cohorts, to more fully appreciate the range of biological variability in the neurotoxic properties of circulating EVs in AD, with the goal of establishing them as functional biomarkers for the disease.

Tracking EVs in the various compartments where they appear (extracellular fluid, peripheral blood) in vivo, is extremely challenging, particularly for human studies, given that there are no effective techniques determining their rates of secretion and uptake. Therefore, it is also challenging to determine the physiologically relevant ratio of vesicles per cell. We base the ratio used in this study on the fact that the conditioned media of 15 × 10^6^ cells contain about 50 × 10^8^ EVs (by NTA of EVs isolated via ultracentrifugation with efficiency of recovery < 30%). CSF contains about 10^8^–10^9^ EVs/mL based on NTA (Supplementary Figure 6C in [46]). Erring on the side of caution and to avoid over-estimating any effects, we selected a lower-end concentration of EVs to be used as treatments to the cultured neurons and took care to keep this consistent between treatments (2.7 × 10^8^ EVs/mL; 600 EVs/cell for 100,000 cells in a final volume of 220 µL).

In these studies, we did not follow cells for more than 48 h after treatment with EVs and, therefore, cannot speak on the potential for long-term changes (e.g., the possibilities that these neurons might recover or proceed irrevocably to degeneration). The present proof-of-principle study intends to motivate a line of research that will address these among similar questions. Future experiments should explore the potential range of effects for a wider range of doses, using a broader logarithmic concentration range and longer incubation periods, with and without replenishment of EVs every 24 h.

The pathophysiological relevance of studying the effects of circulating NEVs and AEVs on neurons in vitro, rests on the assumption that some functional properties (including their neurotoxicity) in the two compartments, brain extracellular fluid, and peripheral blood, remain stable, as brain EVs can cross the BBB intact [64]. Therefore, their functional properties in vitro might provide a window into the molecular processes occurring in vivo in the brain in AD, as their cargo reflects the molecular composition of neurons and astrocytes [29,30,65]. Significant gaps of knowledge regarding the origin and fate of circulating NEVs and AEVs in vivo remain, even though studies continue to shed light onto them (e.g., demonstration of AEVs crossing the blood–brain barrier [64], demonstration of enrichment for brain-specific proteins [30,66], and mi-RNAs in NEVs [65]).

Results were consistent with in vitro and in vivo studies showing that de-regulation of the brain complement system underlies synaptic loss in AD, which could be rescued by complement inhibition [7,15], but critically contribute to the role of EVs in the process. For years, the brain was considered an immune-privileged organ. Today, it is known that various brain cells synthesize immune protein components of both the classical and alternative complement pathways, including the factors required for the MAC assembly, essentially, a membrane pore-forming complex mediating cell death through osmotic lysis. Moreover, the MAC endogenous inhibitor CD59 is expressed by neurons [67] and astrocytes [67,68], likely to protect them from uncontrolled neuroinflammation [69,70]. CD59 deficiency causes catastrophic neuroinflammatory conditions [71,72,73]. In AD, protein levels of CD59 and other complement inhibitors are diminished, whereas those of the complement effectors increased [70,74], rendering neurons susceptible to complement attack [75]. The pathway leading to MAC formation in the AD brain remains unknown, and our findings did not favor the classical, alternative, or lectin pathways. Whichever the pathway, our findings provide proof of concept that EV uptake by neurons might be involved in MAC formation and neurotoxicity.

Recent evidence challenges the notion that neuronal loss is universally detrimental in AD, since removal of damaged neurons might improve neuronal circuit function and functional outcomes [76]. This raises the possibility that complement-mediated neuroinflammation might initially be a compensatory mechanism in the face of AD pathology, by removing misfolded proteins, cellular debris, and inactive synapses, and enhancing plasticity, neural repair, and neuroprotection [77]. However, in vitro and animal model studies show that persistent inflammation in AD becomes maladaptive, resulting in neurotoxicity and synapse loss [3,7,78,79,80].

Neuronal death in neurodegenerative diseases is thought to occur via two main pathways—apoptosis and necrosis [81,82,83]. Neuronal death via apoptosis is characterized by cell shrinkage, chromosome condensation, and DNA fragmentation, incited by either extracellular and intracellular signals that are precipitated by caspases. Apoptosis is often accompanied by the formation of plasma membrane blebs that are either phagocytosed by neighboring glia or degraded within lysosomes. On the other hand, neuronal death via necrosis is characterized by loss of cell membrane integrity and leakage of intracellular contents, and does not involve caspase activity. In the present, the best characterized pathway of programmed necrotic cell death is termed necroptosis, which is initiated by the activation of death receptors, like for example TNF alpha receptor 1. The activation of death receptors is followed by the translocation of RIP1 from the membrane to the cytosol, where it interacts with the necrosome complex and mediates the phosphorylation of MLKL (known as the ultimate effector of necroptosis) leading to membrane disruption. Hence, detecting the protein levels of phosphorylated MLKL is useful for identifying necroptosis activation [84]. In accordance with in vitro, in situ, and post-mortem AD studies showing that necroptosis is implicated in neuronal death in AD [84], we demonstrated that AEV-mediated neurotoxicity proceeds through membrane disruption, in association with the phosphorylation of MLKL.

Although the amyloid hypothesis was the predominant theory for AD pathogenesis [85], efforts aimed at translating it into effective therapies failed, raising renewed interest in alternative therapeutic targets, particularly neuroinflammation. Our results suggest that inhibition of neurotoxic complement components and complexes or of their trans-cellular transfer through EVs might be therapeutically beneficial in AD. Complement inhibitors are already under development for various clinical applications—for example, eculizumab is a well-tolerated monoclonal antibody that inhibits C5 and is being used in paroxysmal nocturnal hemoglobinuria [86], whereas a monoclonal antibody against C1q is well-tolerated in healthy volunteers and is being subjected to a phase I clinical trial (Clinicaltrials.gov number: NCT03010046). C3 inhibitors were tested in non-human primate models of periodontitis and hemodialysis inflammation, with promising results [87,88]. Furthermore, oral administration of a C5 receptor inhibitor decreased pathology in AD mice [16]. However, there are fundamental drawbacks for using systematically administered inhibitors in AD, since they might not easily cross the BBB and might also result in immunosuppression. Endogenous or engineered EVs might be used to overcome the BBB [89] and effectively deliver naturally occurring or synthetic complement inhibitors. The present study demonstrated that circulating AEVs and NEVs are readily uptaken and produce robust effects, motivating the use of modified AEVs and NEVs in brain-targeting therapies for AD.

## Figures and Tables

**Figure 1 cells-09-01618-f001:**
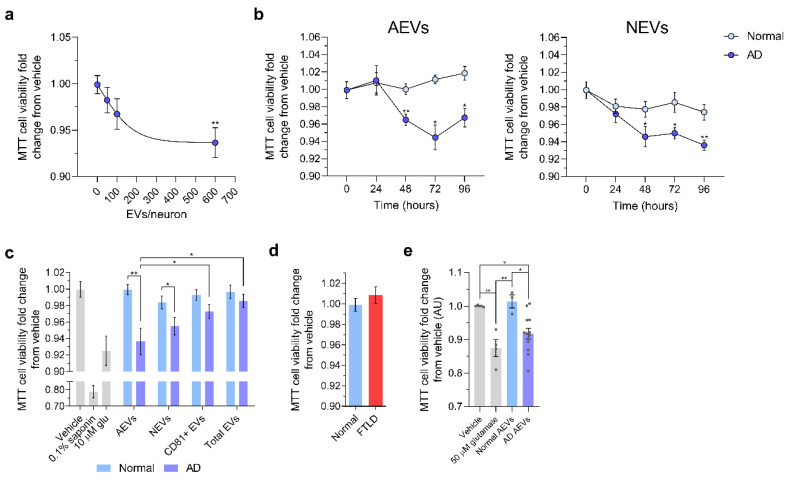
Alzheimer’s disease (AD) astrocytic-origin extracellular vesicles (AEVs) and neuronal-origin extracellular vesicles (NEVs) impair neuronal viability. (**a**) Scatter dot plot showing the MTT cell viability fold-change from vehicle of E18 rat cortical neurons treated with AD AEVs in a concentration-dependent manner. Each dot represents the mean value ± the standard error obtained from neurons treated with plasma-derived AEVs from 7 AD participants at the indicated concentration or vehicle (0 EVs/neuron) in triplicates. Trendline shows the four-parameter logistic nonlinear regression. ** *p* = 0.0015, 600 vs. 0 EVs/neuron; one-way ANOVA corrected for multiple comparisons using the Dunnet test. (**b**) Scatter dot plots with the connecting line showing the MTT cell viability fold-change from the vehicle of E18 rat cortical neurons treated in triplicates with AEVs or NEVs from 7 AD participants and 6 normal controls at 600 EVs/neuron, in a time-dependent manner. Each dot represents the mean value ± SEM. AD vs. Normal: AEVs, 48 h ** *p* = 0.0015, 72 h * *p* = 0.0191, 96 h * *p* = 0.0201; NEVs, 48 h * *p* = 0.0337, 72 h * *p* = 0.0127, 96 h ** *p* = 0.0016; two-tailed unpaired t-test with a confidence interval of 95%. Time-dependent EV treatment vs. vehicle treatment (0 h): AEVs, 48 h ** *p* = 0.0060, 72 h **** *p* < 0.0001, 96 h ** *p* = 0.0023; NEVs, 48 h **** *p* < 0.0001, 72 h * *p* = 0.0186; 96 h ** *p* = 0.0028; ordinary one-way ANOVA. (**c**) Bar graph showing a significant decrease in MTT cell viability (fold-change from vehicle) in E18 rat cortical neurons treated with AEVs and NEVs, but not CD81+ EVs and total plasma EVs from 7 AD participants, compared to neurons treated with AEVs and NEVs from 6 normal controls, respectively. Neurons treated with 0.1% saponin detergent and 10 μM glutamate were used as positive controls for neurotoxicity. Each bar represents the mean value ± SEM from triplicate wells and five different experiments; the condition of 600 EVs/neuron and incubation for 48 h was selected based on **a**, **b**. Analysis was based on mixed linear model accounting for technical (well, experiment) and biological (Subject, Group) variability and including all EV types. For NEVs, AD vs. Normal, *p* = 0.003. For AEVs, AD vs. Normal, *p* = 0.006. For CD81+ EVs, AD vs. Normal, *p* = 0.904. For total EVs, AD vs. Normal, *p* = 0.675. For AD participants, NEVs and AEVs had similar effects on MTT (AEVs vs. NEVs, *p* = 0.695), but AEVs decreased MTT compared to both CD81+ (*p* = 0.035) and total EVs (*p* = 0.029), and NEVs showed similar strong trends in pairwise comparisons (*p* = 0.087 and *p* = 0.067 respectively). * *p* < 0.05, ** *p* < 0.01. (**d**) Bar graph showing the MTT cell viability fold-change from the vehicle of neurons treated with AEVs from two FTLD and 6 normal control participants. No difference in cell viability was observed (*p* = 0.934). Treatments were carried out as indicated in (**c**). Bar graph showing a significant decrease in MTT cell viability in E18 rat cortical neurons treated with AEVs from 13 autopsy-confirmed AD patients and 3 neurologically normal controls from JH ADC, all part of an independent cohort different from that used for (**a**–**e**). Each bar represents the mean value ± SEM from five wells and two independent experiments. Vehicle vs. 50 µM glutamate, ** *p*= 0.0038; Vehicle vs. AD AEVs, * *p* = 0.0225; 50 µM glutamate vs. Normal AEVs, ** *p* = 0.0048; Normal AEVs vs. AD AEVs, * *p* = 0.0225.

**Figure 2 cells-09-01618-f002:**
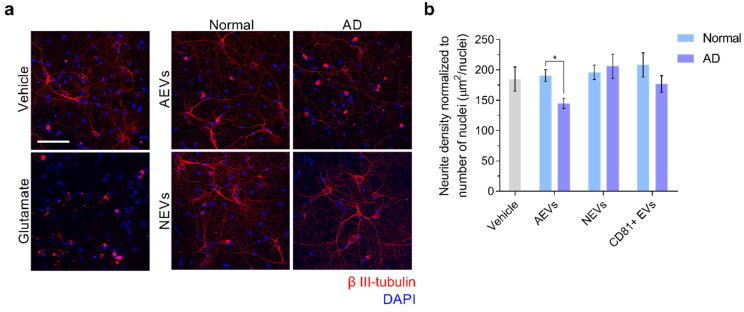
AD AEVs induce neurodegeneration. (**a**) Immunofluorescence of β III-tubulin (red) as an indicator of neurite density in E18 rat cortical neurons incubated with either AEVs or NEVs isolated from the plasma of AD patients and normal controls at 600 EVs/neurons for 48 h. Fluorescent confocal microscopy images at 25X magnification show that neuronal cultures treated with AEVs from AD participants exhibit decreased neurite density, compared to the vehicle-treated neurons and neurons treated with AEVs from normal participants, as well as with NEVs from AD and normal participants. Treatment with 100 μM glutamate was used as a positive control for neurotoxicity. DAPI staining in blue shows the localization of nuclei. Scale bar, 100 μm. (**b**) A bar graph showing the quantification of neurite density defined as the image area covered by β III-tubulin signal above the threshold normalized to the number of nuclei. Each bar represents the mean value ± SEM of 5–15 images, per treatment with NEVs, AEVs, and CD81+ EVs from 4 AD subjects and 3 normal participants in duplicates. There was significant neurite density decrease in the neurons treated with AEVs from AD compared to the normal participants (AD vs. Normal, *p* = 0.02); the difference was not observed in the NEV-treated (*p* = 0.891) or CD81+ EV-treated neurons (*p* = 0.252), suggesting a greater neurodegenerative potential for AEVs, compared to NEVs or CD81+ EVs. * *p* < 0.05.

**Figure 3 cells-09-01618-f003:**
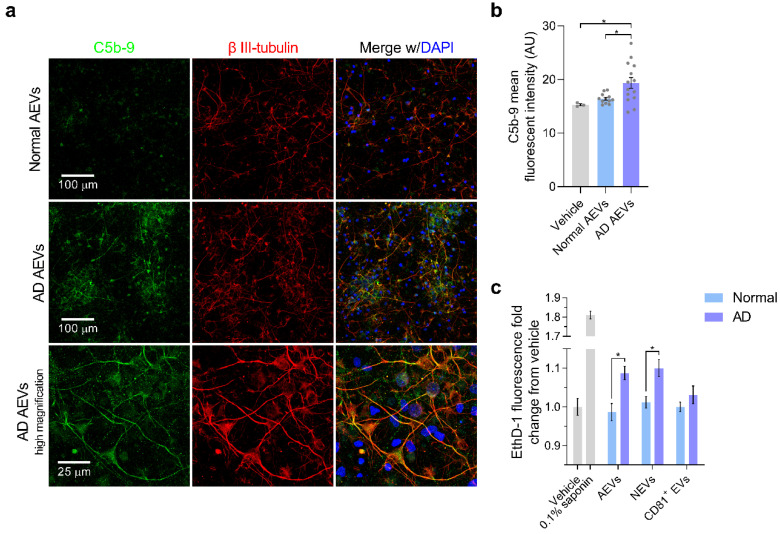
The AD AEV-mediated accumulation of the complement membrane attack complex in neurons is associated with membrane disruption. (**a**) E18 rat cortical neurons were incubated with AEVs isolated from the plasma of 2 AD patients and 2 normal controls at 600 EVs/neuron for 1 h. After incubation, neurons were subjected to C5b-9 immunofluorescence (green), with β III-tubulin (red) co-staining as a marker of neuronal soma and processes. DAPI staining in blue shows the localization of nuclei. Fluorescent confocal microscopy images at 20X magnification show an increased accumulation of the complement membrane attack complex in the neural soma and the neurites of cultures treated with AD AEVs, in comparison with cells treated with AEVs from normal controls and the vehicle (AD vs. Normal, * *p* = 0.026; AD vs. vehicle, * *p* = 0.012; Normal vs. vehicle, *p* = 0.051). In the bottom panel, an image at 63X clearly shows the neuronal distribution of C5b-9 co-localizing with β III-tubulin. Scale bars: 20×, 100 μm; 63×, 25 μm. (**b**) Quantification of (**a**) A bar graph with individual values in grey shows the C5b-9 mean fluorescence signal ± SEM obtained from 3–5 images of neuronal cultures treated with AEVs from the plasma of 2 AD patients and 2 normal controls in duplicate. Treatment with vehicle was used to determine the basal detection of C5b-9. (**c**) E18 rat cortical neurons were incubated with AEVs, NEVs, or CD81+ EVs isolated from the plasma of 4 AD and 4 normal control participants at 600 EVs/neuron for 48 h. After incubation, neuronal cultures were subjected to the fluorescent EthD-1 membrane disruption assay. Treatment of neurons with NEVs and AEVs from 4 AD compared to 4 normal control participants, resulted in a significantly increased EthD-1 fluorescence fold-change over vehicle (For NEVs, AD vs. Normal, *p* = 0.044; for AEVs, AD vs. Normal, *p* = 0.024), suggesting that AD NEVs and AEVs mediate neuronal membrane disruption; there were no differences in EthD-1 fluorescence in neurons treated with CD81 + EVs (*p* = 0.332) from AD, compared to normal control participants. Neurons treated with 0.1% saponin detergent were used as a positive control of neurotoxicity with membrane disruption. Each bar represents the mean value ± SEM of EthD-1 fluorescence fold-change from vehicle treatments with EVs from 4 AD and 4 normal control participants in triplicates, carried out in three different experiments. * *p* < 0.05, ** *p* < 0.01.

**Figure 4 cells-09-01618-f004:**
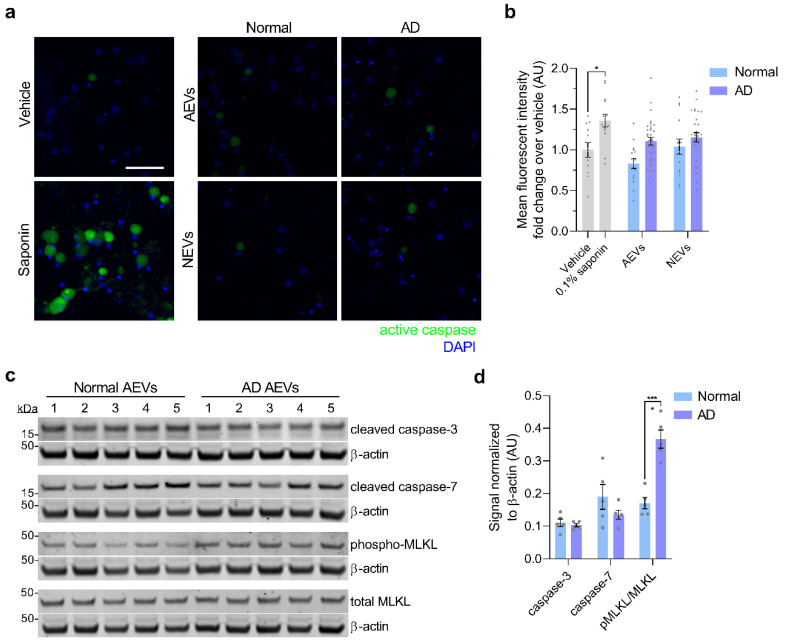
Neuronal activation of necroptosis, but not apoptosis, by AD AEVs. (**a**) Neurons treated with AD AEVs were subjected to an in vivo fluorescent caspase activity assay that allows the fluorescent detection of active caspases in live cells (green). Treatment with 0.1% saponin detergent was used as a positive control for caspase activation. The membrane permeable nucleic acid dye Hoechst 33342 was used to visualize nuclei (blue). Scale bar, 100 μm. (**b**) Quantification of (**a**) A bar graph of the quantification of the mean fluorescent intensity in ROIs selected upon thresholding, using the Fiji-Image J max entropy algorithm on confocal microscopy images shows that apoptosis is not active in neurons incubated with AEVs and NEVs from 4 AD, compared to 4 normal control participants in duplicate. Each bar represents the mean value ± SEM of 5–10 images per treatment. Statistical analysis: One-way ANOVA corrected for multiple comparisons using the Dunnett test; * *p* = 0.0282. (**c**) Western blots against cleaved caspase-3 and -7 as markers of apoptosis, and relative abundance of phosphorylated over total MLKL as a marker of necroptosis. β-actin was used as the loading control. (**d**) Quantification of (**c**) showing a significant increase in the relative abundance of phosphorylated MLKL over total MLKL, normalized to β-actin, in rat cortical neurons treated with AD and normal control AEVs. No differences in cleaved caspase-3 and -7 were observed. Each bar represents the mean value ± SEM of the protein optical density normalized to β-actin in neuronal lysates after treatment with plasma-derived EVs from 5 AD and 5 normal control participants in triplicates. Statistical analysis—one-way ANOVA corrected for multiple comparisons using the Dunnett test; **** *p* < 0.0001.

**Figure 5 cells-09-01618-f005:**
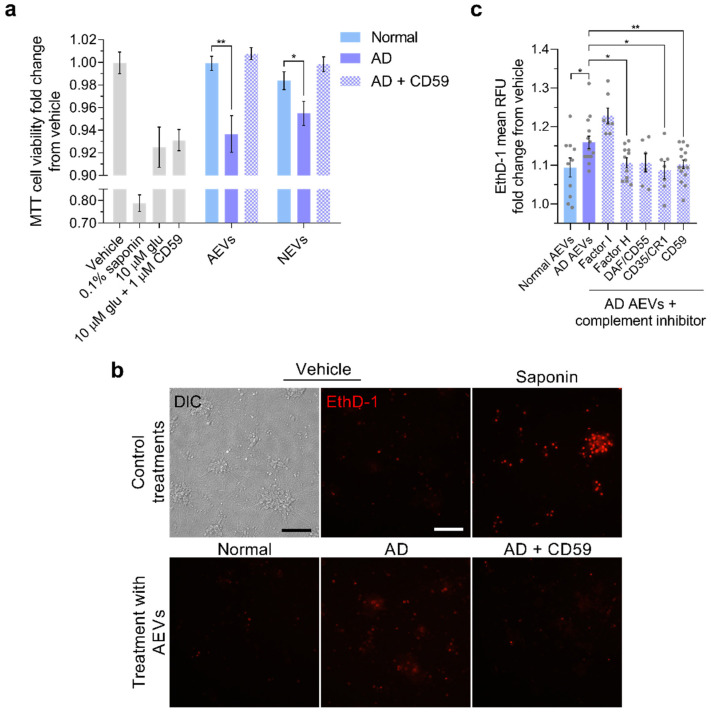
Complement inhibition prevents the neurotoxicity of AD AEVs and NEVs. (**a**) Bar graph showing significant decreases in MTT cell viability (fold-change from vehicle) observed in E18 rat cortical neurons treated with AEVs and NEVs from 4 AD participants, compared to neurons treated with AEVs and NEVs from the same 4 AD participants co-incubated with MAC inhibitor CD59 and compared to neurons treated with AEVs and NEVs from 4 normal controls; 600 EVs/neuron for 48 h were used. Neurons treated with 0.1% saponin detergent and 10 μM glutamate were used as positive controls for neurotoxicity. Co-treatment of neurons with 10 μM glutamate and 1 μM CD59 shows that CD59 does not exacerbate the glutamate-mediated cell viability impairment nor does it rescue it. Each bar represents the mean value ± SEM obtained from the neurons treated in triplicate wells in two individual experiments. For NEVs, AD vs. Normal, *p* = 0.013; AD plus CD59 vs. Normal, *p* = 0.859; AD vs. AD plus CD59, *p* = 0.113. For AEVs, AD vs. Normal, *p* = 0.001; AD plus CD59 vs. Normal, *p* = 0.609; AD vs. AD plus CD59, *p* = 0.016. (**b**) Human iPSC-derived neurons incubated with AD AEVs at 800 EVs/mL for 48 h showed an increased nuclear EthD-1 fluorescence, in comparison with neurons treated with AEVs from normal controls and vehicle-treated cells. Co-incubation with CD59 blocks the AD AEV-mediated EthD-1 fluorescence increase. Neurons treated with 0.1% saponin detergent were used as a positive control of neurotoxicity. Scale bar, 200 μm. (**c**) Bar graph with individual values in grey showing the EthD-1 mean relative fluorescence units (RFU) fold-change over vehicle ± SEM from human iPSC-derived neurons treated with AEVs from 5 AD participants, with or without the indicated complement inhibitors and 5 normal controls in duplicates or triplicates. * *p* < 0.05, ** *p* < 0.01, ***** *p* < 0.0001.

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
