# Peer review of "Astrocyte- and Neuron-Derived Extracellular Vesicles from Alzheimer’s Disease Patients Effect Complement-Mediated Neurotoxicity"

_cells, 2020, doi:10.3390/cells9071618_

Round 1

Reviewer 1 Report

The paper explores the potential of extracellular vesicles originating from two types of cells, i.e. neurons and astrocytes, isolated from plasma samples from AD cases [CDR 0.5-1 (questionable or very mild to mild) – NIA-AA MCI due to Alzheimer’s disease/high likelihood of AD – IWG-2 typical prodromal Alzheimer’s disease cases] or controls to induce in vitro toxicity, involving metabolism, neuronal remodelling, and finally necroptosis against neuronal cells. The authors also proposed that Complement components play a role in the process.

In contrast to a previous study of the group that used CSF or plasma EVs to study in vitro mitochondrial respiration and apoptosis, this work further focuses on the subtypes of EVs and on a different mechanism of action. The work is interesting; however, it suffers many of the problems of a multidisciplinary project:

1) The initial sample (human material) is limited, but precious. Classification is also quite confusing in establishing a uniform cohort, probably because of the fluency of guidelines. IWG-2 criteria were developed to “allow earlier intervention in the prodromal stages of the disease and to facilitate research studies into secondary prevention of AD in the preclinical states” [Dubois B et al, 2014]. Furthermore, an AT(N)(C) staging has been proposed [Clifford R Jack Jr et al, 2018], and both NIA-AA & IWG-2 may contribute [Molin & Rockwood, 2016]. Therefore, it would be essential to consider patient data about any available “biomarker” in the experimental procedure and data analysis. Unfortunately, this kind of crucial information has been moved to the supplementary materials, and it seems that there is a mixed-up with patient data, see specific comments.

2) The authors did a good work regarding the presentation and sorting of the main biological material used herein - microvesicles, following the recent nomenclature. However, there is no clear idea and perspective about EVs production, content, trafficking, and fusion. Adding to this picture that CNS is shedding material to the periphery [Shi M et al, 2019], the picture is getting more complicated, and there are points in the Discussion, where even a transport back to CNS is proposed. The presented ideas and evidence are not supporting a compartmentalized model, where each cell/tissue/organ has some degree of independence and specific physiological function, and a cross-talk, or loop signalling (?), may facilitate or aggravate these functions.

3) Dosing is very tricky. Usually, we work on a log scale. There is a confusion about how much EVs should be used, and are used per cell. In my opinion, the maximal effect, about 10% (?) is not considered essential. Most importantly, the authors provided a series of EVs features from AD/control subjects, where there are essential differences in number or quality, besides the previously published Aβ content. I believe that this basic information should shape the experimental course, patient selection, dosing, etc.

4) Similarly to the loose idea about neuronal, and astrocytic function, a far-fetched idea on necroptosis appeared. Tuj-1 is a well-established marker of neuronal commitment and differentiation, and its cellular localization is crucial as well in microscopy experiments [Fanarraga ML, Avila J, Zabala JC, 1999; Neural Stem Cells - Methods and Protocols, Editors: Weiner, Leslie P, 2008].

5) MPLK appearance may be controversial or closer to regeneration, and I get a completely different picture from the same literature sources [Yoon S et al, 2017; Ying Z et al, 2018, commented in Vandenabeele P et al], and the Figures of this manuscript. It is necessary to check ref, and ab against different epitopes, or perform new experiments using new reagents.

6) Moving on to the immune response. Many pathways have been proposed by the authors, and it is not clear whether innate or adaptive immunity, or the alternative pathway are in focus. Ref to C5a are used, but there is no direct link between this molecule and MAC. Furthermore, the panel used (Luminex kit HCMP2MAG-19K) does not include MAC relevant analytes. The series of experiments is diverse, and does not support any discrete mechanism.

There is an interesting study on the detection and localization of C5b-9 in the periphery [Yell PC, Diffuse microvascular C5b-9 deposition is a common feature in muscle and nerve biopsies from diabetic patients, 2018], while many other studies on immune partners and EVs’ content and function should be replaced.

7) Finally, there is some mention to cell variability in the abstract/introduction which could be misleading, and it is not justified, or discussed. Variability in the nervous system is described and assessed at different levels, cytometry, asymmetry, phenotype, response to stimuli. Accordingly, neurodegeneration is a wide umbrella [Przedborski S et al, 2003; Yaron & Schuldiner, 2016], and various parameters should be considered, upon reaching conclusions. Regarding the differentiation of neuronal phenotype, this paper could be also of use [https://elifesciences.org/articles/50333].

In conclusion, I found the idea of peripheral EVs use exciting. However, the submitted manuscript contains dispersed information, and there is definitely a lack of physiological concept, and mechanistic focus. I would propose to the authors to spend some time revisiting literature, and then undertake a more concrete plan, not necessarily exhaustive, to take benefit of their precious material.

Back to the trivial points, there is a need to check M&M sentences format, numbers, reagents, manufacturers/suppliers full details, and reformat the literature as well, and pay some attention to capital/small characters.

Here there is a list of suggestions, mainly to the executive part.

iPSC

L27 “were not produced” but “were supressed”?

L28 MAC deposition (?). How it has been explored?

L32-L34 Please provide a reference

L52 in the hippocampus, AND temporal and frontal lobes of AD patients

L58-69 Up-to-date and focused references could be used

L72 and to ensue/ensuing (?)

L86 Is it GDR global score 0.5-1, or as described in Table S2 6±0.2?

L121 Please note vector’s and Polybrene’s final concentration.

L126 Please note the final concentrations for BDNF, and NT3.

L136 complied

L139 Rephrase: thrombin catalyzes the conversion of fibrinogen to fibrin

Technical question: clots are supposed to entrap molecules/complexes and they do so for vesicles as well [Arakelyan A et al, 2016]. Did this step result in low yield or bias?

L142 protease/phosphatase inhibitors cocktail, please cite the manufacturer

L144 Define ExoQuick kit. There is a mention for patent application, but all procedures are quite known.

L143-5 i) Please rephrase procedure. ii) There is an error persisting through literature: 252 μl (?) iii) did the authors perform a one-step collection or what?

L157-9 0.1 M glycine, pH ? 1M Tris ? pH ?

L159 Ten

L162 M-PER Mammalian Protein Extraction Reagent

L164 Only AEVs were checked?

L159-168 Please rephrase. Use short sentences and regroup procedures.

L166-167 Mesh coat – material – full name, uranyl concentration, supplier?

L175 Bis-Tris

L177 using the iBlot2; full details at each step.

L182 Golgi

L185 5x wash? Isn’t exhaustive? Is it necessary?

L186 TBS is supposed to be explained earlier, then 0.05% Tween-20 TBS

L181-7 First, these lines are confusing. It is not clear what was the purpose of the experiment. Did the authors would like to confirm isolation, purity, or what? Please state the aim, then classify markers, and then provide references.

L188 It is better to split the assays into specific protocols, or to individual paragraphs.

L195 assessed using the membrane impermeable dye ethidium homodimer-1 (EthD-1) (Cell Biolabs, San Diego, CA), which fluoresces when bound to DNA.

L196-200 i) Understood, but there could be some equivalent, like ~100 EVs per neuron, or/and estimation of initial serum volume, and in ref#40 100 EVs/neuron were used. In the next paragraph, details are given!  ii) It would be also useful to know if there were quantitative and qualitative differences among cases, and how they were incorporated.

L202 Is there any reason to work at 100% confluency?

L206 ddH2O/ultrapure/etc, please use uniform terms throughout.

L207 Glutamate has been used in various concentrations, and in ref#40 at 100 mM. It is possible to use at concentrations comparable to the range of expected effect, but there are concerns about the per se effect as described herein and elsewhere.

L208 & later in L304 Saponin (of what quality) was used? Was it used as a positive control for ethidium homodimer-1 assay? It is not exactly a reagent inducing neurotoxicity, many other compounds could be used instead. Ref.

L210-212 Please rephrase. AND L212-233, there is a duplication of M&M, why a validation was needed? Did an observer record the number of positive cells? Did an observer used some sort of software?

L217 It is unclear why CD59 was used prior to experiments, in all cases (?). Again, please follow common sense when describing your experimental set-up, and provide references for each condition.

L221-226 Again, there is some confusion about the ratio of EVs per cell. If 0.5 ml was used per well, a final ratio of 750EVs/neuron would be expected. In addition, why the highest ratio (50, 100, 600 in L203) was used in this experiment?

L228-235 How EC50 was established? Ref, suppliers.

L242 Tuj-1, ref.

L258 Mounting medium (?).

L291-292 Please rephrase

Reviewer 2 Report

Article title “Astrocyte- and neuron-derived extracellular vesicles from Alzheimer’s disease patients effect complement-mediated neurotoxicity” by Nogueras-Ortiz et al is well planned and carefully designed study. The Paper is well written with the elaborative method section. This paper improves our current understanding of the increasing role of EVs in several neurological disorders. Authors found that AEVs and NEVs are neurotoxic and work through induction of membrane attack complex (MAC) in rat cortical neurons and human iPSC-derived cell line. I think the experiment and data are supporting their hypothesis. Although authors have presented several lines of evidence for AEV/NEVs mediated toxicity and their recovery after inhibition of involve pathway in cells but the level of toxicity is a major concern. Are this much level of toxicity is physiologically relevant even though it partially matched with glutamate-mediated toxicity?

  Some of the minor concerns are mentioned below  

  1. I would suggest using visibly distinguishable bullet color in fig 1b. It appears hard to make out with overlapping colors in graphs.
  2. Line 142; Include details for protease/phosphatase inhibitor (like Cat. No. etc.)
  3. Line 157; Includes pH for Glycine buffer.
  4. Section 3.1; Mentioning the Information about the size of EVs will be helpful.
  5. Figure S1e; Purity of isolated EVs was evaluated by showing the WB for different marker proteins present on EVs from control participants. It would be interesting to see if there is any difference observed in the marker protein expression level in EVs isolated from AD participants. I will suggest to include WB from AD as well in the supplement.
  6. Figure 1a, b; Comparing the pattern of cell viability among AEVs and NEVs, it is apparent that AEVs treated cells are recovering after 48h whereas NEVs have a declining trend. It is also evident from the P-value at 48 hr and 96 hours for AEVs and NEVS respectively. It could have been better to use a different time point for AEVs and NEVs considering their variable effectiveness at different time points.
  7. Figure 2a, Since NEVs demonstrated a more significant effect at 96 hr (fig 1b), it is likely, we may see a greater effect in neurite density in samples treated with NEVs if the experiment could have been done at a later time point.
  8. Line 687; mention composition of “buffer C”.
  9. Line 708; use number for volume used.

Reviewer 3 Report

This is a well-written manuscript providing a thorough investigation of astrocytic extracellular vesicles as a mechanism of transmission for complement-mediated neurodegeneration in Alzheimer’s disease.  I have none but minor editorial concerns with this paper which I read with great interest.

Please define the iPSC at first mention – as written, the term is first defined in line 527

In Figure 3 (line 470):  Particularly in this legend, but true in all legends, it is difficult to separate the figure identifiers (a, b, etc) from the text.  It would be helpful to put figure identifiers in parentheses in addition to bolding so they are clearly separated from the text.  In this figure, because scale bars are not clearly different (only the legend indicates differences in magnification), the magnified images of Fig 3a are not clear.  I suggest marking the scale bars to make this clearer (despite the y-axis label ‘high magnification’) – the y-axes of all three image rows could include the magnification or it could be included with the scale bar. Finally, Fig 3c labels are not noted in the legend (line 485)

Round 2

Reviewer 1 Report

Although the authors addressed the points underscored and made some corrections accordingly, all major points have not been under consideration, and it is definitely not a matter of my personal opinion.

1) Underscoring the ATNC staging was first due to the .X format in supp Table 1 of the previous version (the dot was not so visible in a small screen), while the rest of the biochemical markers could be questionable, especially because of the lack of control group values [indicative references https://www.ncbi.nlm.nih.gov/pmc/articles/PMC2759394/

https://www.scielo.br/scielo.php?pid=S1516-44462019005009103&script=sci_arttext

https://www.sciencedirect.com/science/article/pii/S2352873717300513]. The inclusion of units is welcome as well.

2) Most importantly, the quantifiable differences among the main material (EVs) are of value, and should be considered. That points to the formulation of working hypotheses. I am still struggling to figure them out.

- What is the tissue-target of circulating NEVs, and AEVs?

- Is this speculation physiologically relevant?

- What data (experimental or other) support such a hypothesis?

In my mere opinion, the first possible target of EVs is expected to be circulating cells, then endothelial cells, then organs/tissues.

Is this a model of study of brain neuronal population?

None of these points were considered.

3) To confirm a hypothesis, experimental conditions should conform with other groups, and when data are not available, with physiological data, and there are many of both categories:

- The use of a neuronal population, as a possible target is not justified, as explained above.

If it is justified:

- Is the ratio of vesicles per cell physiologically relevant? Condition-relevant?

- Experimental setup of neuronal cultures is beyond ideal. Again, the end does not justify the means. MTT is working for a range of situations and applications, and it is quite peculiar why it was difficult to produce quantifiable results. It is also not wise to work within a 5% viability window, and convince anyone to keep looking for differences.

- Cultures should not start at confluency, especially when neurons are expected to grow and differentiate. The profile of the target population is not as expected, what has been described decades ago, and recently for iPSCs [https://www.nature.com/articles/s41598-017-12452-x].

- Differences in number and morphology of cells are expected, but no effort to understand the behaviour of cells in culture over a period of time has been made.

4) Therefore, it is no surprise that various problems arose subsequently upon introducing various insults to these cultures, and why the effects are marginal.

I am still trying to make some sense of the data, mainly because of the scarcity of the human material.

I would advise authors to make a proper use of the literature regarding cell death [Galluzzi 2018 Molecular mechanisms of cell death Recommendations of the Nomenclature Committee on Cell Death 2018; Tang 2019 The molecular machinery of regulated cell death], and the previous citations.

In addition, in Figure 4, it appears that there was some caspase activity, but the blots point to caspase-3 and caspase-7 inhibition (?). I would advise to re-analyze the blots/plots again. Taken together with p-MLKL, it could support the activation of a pathway, and some complement components could further help the discussion. To do so though, we have to get rid of the activity assay, because the microscopy technique is not suitable (we don’t want to change image threshold), and understand differentiation-RCD-survival, remove all introductory and final comments on apoptosis etc.

Similarly, a critical integration of immunology relevant results could be of use, while I don’t follow the idea of their omission, as elaborated in authors’ response.

In conclusion, this work needs fundamental reshaping. I was hoping, that my initial comments would be helpful in improving the understanding of the situation, and the perspective, and then move to some minor changes. At this moment, there is no progress.
